# Connecting Health and Technology: Validation of Instant Messaging for Use as Diabetes Mellitus Control Strategy in Older Brazilian Adults

**DOI:** 10.3390/ijerph22020282

**Published:** 2025-02-14

**Authors:** Manoela Vieira Gomes da Costa, Renata Puppin Zandonadi, Verônica Cortez Ginani, Silvana Schwerz Funghetto, Luciano Ramos de Lima, Tania Cristina Morais Santa Barbara Rehem, Marina Morato Stival

**Affiliations:** 1Graduate Program in Health Sciences and Technologies, Campus Universitario Ceilândia, University of Brasília, Brasília 72220-275, Brazil; silvanasf@unb.br; 2Department of Nutrition, Faculty of Health Sciences, Campus Universitario Darcy Ribeiro, University of Brasília, Brasília 70910-900, Brazil; renatapz@unb.br (R.P.Z.); vcginani@unb.br (V.C.G.); 3Department of Nursing, Ceilândia University Campus, University of Brasília, Brasília 72220-275, Brazil; ramosll@unb.br (L.R.d.L.); taniarehem@unb.br (T.C.M.S.B.R.)

**Keywords:** aging, type 2 diabetes mellitus, glycemic control, nursing, nutrition, social media, public health

## Abstract

Background: Digital technologies are increasingly being used to promote effective health interventions in the older adult population. This study aimed to develop and validate instant messages with educational content related to glycemic control, delivered via an instant messaging application, as a type 2 DM control strategy for older adults. Methods: This was a methodological study that developed instant messages containing text and images. The validation process was conducted by a panel of experts composed of nursing, nutrition, and physical education professionals. SPSS version 25.0 was used for the statistical analysis. A CVI was used to measure the experts’ agreement regarding the validity of the content of the educational messages. An exact test of binomial distribution with *p* > 0.05, indicating statistical significance, and a 0.95 proportion of agreement was used to estimate the statistical reliability of the CVI. Results: Sixty-one text messages were prepared with illustrations as support, and were divided into three themes. The educational messages were validated by experts, with an average CVI score above 0.80 for all indicators. Conclusions: The educational messages developed in this study were considered relevant and clear for older adults and could be applied in the digital environment, with the objective of helping older adults manage type 2 DM.

## 1. Introduction

In Brazil, population aging is a reality. In 2022, 15.6% of the Brazilian population were older people [1]. With the increase in life expectancy and the process of globalization, which encourages a sedentary lifestyle and poor eating habits, individuals have become more likely to develop chronic non-communicable diseases (NCDs), including diabetes mellitus (DM) [2,3].

Diabetes mellitus (DM) is a chronic disease characterized by increased blood glucose (hyperglycemia). This increase occurs when the body does not produce a sufficient amount of insulin, leading to type 1 diabetes mellitus, or when the body is unable to effectively use the amount of insulin produced by the pancreas, leading to type 2 diabetes mellitus (DM2) [4].

Hyperglycemia is a chronic condition that can cause serious health problems, including cardiovascular diseases (CVDs), nerve damage (neuropathy), kidney damage (nephropathy), eye diseases that mainly affect the retina (retinopathy), and lower limb amputation [4].

The prevalence of DM is reaching epidemic proportions. Worldwide, in 2021, in the age group of 20 to 79 years, 537 million adults and older adults were diagnosed with DM, and approximately 6.7 million people died from this disease and its complications [4]. In Brazil, DM affected 20 million people, around 10.2% of the population [5]. In the ranking of countries with the highest number of people diagnosed with DM, Brazil was in the 6th place in 2021 for the same age group, generating an expenditure of USD 966 billion [4].

DM treatment aims to prevent and delay micro- and macrovascular complications and optimize patients’ quality of life [6]. To this end, treatment is based on glycemic control [7], which can be achieved by leading a healthy lifestyle, maintaining an ideal body weight, eating an adequate diet, and exercising regularly [8]. It is recommended that functionally independent older people with DM have an HbA1c of <7.5% [9].

Health education for DM is the primary strategy to ensure self-care, placing the individual at the center of care, encouraging people to change their lifestyle, and raising awareness of the risks of this disease if glycemic control is not achieved [10]. Educational actions should arouse in DM patients an interest in understanding how the disease works so that this knowledge can help them in self-care, thus promoting health and well-being [11].

Healthcare activities were affected by the COVID-19 pandemic in 2020, and it became necessary to increase the use of technology to meet healthcare demands, reformulate strategies, and make it possible to execute these strategies online. The use of digital technologies became even more prevalent [12]. The period of social isolation during the COVID-19 pandemic drove older adults to the digital environment [13]. Therefore, the World Health Organization (WHO) proposed the Global Health Strategy 2020–2025, encouraging the use of information and communication technologies to develop digital health. This strategy aims to strengthen healthcare systems to guarantee the health and well-being of the population [14].

Digital and educational technologies are used to promote effective interventions for the health of older adults [15,16]. Several types of educational technologies have been used to achieve better glycemic control in people diagnosed with DM [17,18,19,20,21]. However, most of these educational technologies convey the content in text format, without the use of images to improve understanding.

For a technology to be well accepted by the older population, it must be adapted to their daily needs, be easy to use through simple and intuitive navigation, have appropriate language and an attractive design, and be rich in images that convey everyday life [22]. Among the existing telecommunication modalities, the use of social networking services stands out, with instant messaging applications being well accepted by older adults [23].

In Brazil, 88% of the population aged 10 or older uses the internet. Of these individuals, 98.8% use their cell phones to access the internet. The expansion of internet use has accelerated even more among the older population and in rural areas. In Brazil, 66% of the older population uses the internet [24].

WhatsApp is a widely used instant messaging application that allows real-time interactivity, connectivity, portability, and multifunctionality [25]; it presents features that help meet communication demands, overcome time and space barriers, and maintain contact between health professionals and patients [26]. It is part of the Brazilian lifestyle and is used by 147 million Brazilians, of which 86% use the application daily to exchange text messages and images [27]. Older adults also use this tool to clarify doubts about health care [28].

The use of WhatsApp in health education and the monitoring of DM treatment has been described as an important tool for understanding the disease [29], improving self-efficacy and self-management of DM [30], improving glycemic control [31], encouraging medication adherence [32], and reducing acute complications [33].

Older people still have many limitations regarding digital literacy knowledge and skills; however, they are interested enough to seek possible solutions to these barriers [34]. In this regard, this study aimed to develop and validate instant messages with educational content related to glycemic control, delivered via an instant messaging application, as a type 2 DM control strategy for older Brazilian adults.

## 2. Materials and Methods

### 2.1. Study Design and Ethical Approval

This methodological study developed an educational technology in the form of instant messages containing text and images to be delivered via an instant messaging application (WhatsApp). This study constitutes the initial stage of a randomized clinical trial to evaluate the effectiveness of educational messages to promote glycemic control among older individuals diagnosed with type 2 DM. The project was approved by the Research Ethics Committee of the Ceilândia Faculty of the University of Brasília—CEP/FCE/UNB (4.876.336 CAAE 45733521.0.0000.8093)—and by the Research Ethics Committee of the Foundation for Teaching and Research in Health Sciences of the Health Department of the Federal District—CEP/FEPECS/SESDF (4.980.237 CAAE 45733521.0.3001.5553). The ethical requirements for human research followed the Declaration of Helsinki.

### 2.2. Preparation of Educational Messages

The contents of the educational messages presented knowledge about DM, encouragement of self-care, and interventions necessary for glycemic control, such as physical exercise and healthy eating. The leading national content produced by the Brazilian Association for the Study of Obesity and Metabolic Syndrome [35], the Brazilian Ministry of Health [36,37,38,39,40], and the Brazilian Diabetes Society [10,41,42] was used for reference.

The educational messages were created by members of the Health, Care and Aging Research Group (GPeSEn), comprising professors and postgraduate and undergraduate students from nursing, nutrition, and physical education courses at the University of Brasília (UnB).

After the content was developed, a character named Dora (a nurse) was created to present the messages and support the content. To strengthen the relationship between health professionals and patients, a dialogue resource between Dora and an older adult was used. The name was chosen in reference to Dorothea Elizabeth Orem, a nurse who developed the Self-Care Theory [43].

When preparing the educational messages, aspects related to language, images, illustrations, and layout were considered. Simple language was used so that it would be appropriate to the socioeconomic and cultural context, suitable for the target audience, and consistent with the messages to be transmitted. Since the target audience is older adults, the vocabulary used was easy to understand, with short sentences to facilitate understanding at home and encourage self-care [44,45].

When selecting the illustrations, racial and ethnic aspects were considered to target people from a wide range of cultural groups and ethnicities. The illustrations were always placed close to the text to which they referred [45], and most of them were created by a graphic design professional. The messages and illustrations were prepared in the Canva application, with Open Sans as the font and font sizes of 20 and 22 used for the text. The messages were created and validated in Portuguese, and after validation, they were translated to English by experts. Please refer to Appendix A, which contains the instant messages in English.

### 2.3. Content Validation by Experts

The validation of the content of the educational messages occurred in December 2023 using a group of experts recruited through different strategies, including advertisement on the Lattes Platform of the National Council for Scientific and Technological Development (CNPq), on the websites of the leading Brazilian universities, and through referrals of other experts by those who had already participated.

The inclusion criteria were as follows: being a professional in physical education, nursing, or nutrition; voluntarily agreeing to participate in the research; answering all the research forms; and achieving a score of >5 points according to the Fehring criteria [46]. The points were distributed as follows: master’s degree (4 points); master’s degree with dissertation in the area of interest (1 point); research in the area of interest (2 points); published an article in the area of interest (2 points); doctorate with thesis in the area of interest (2 points); professional experience in the area of interest (2 points); and specialization in the area of interest (2 points).

The invitation letter was emailed along with the research objectives and a Google Drive link that allowed access to the Free and Informed Consent Form (FICF), the educational messages, and the validation instrument. The experts signed the FICF electronically and received a copy signed by the researchers via email.

The message evaluation instrument was structured into three parts: (a) characterization by the experts; (b) a validation questionnaire divided into three blocks: objectives, structure and presentation, and relevance; and (c) final evaluation of the messages. The validation instrument was adapted from other studies that validated educational technologies in the health area [47]. A period of 30 days was granted to complete the validation, and if it was not completed, a new email was sent every 7 days to reinforce the invitation.

A 5-point Likert scale was used to measure the level of agreement and disagreement in the experts’ responses. The responses were categorized as follows: 1—I completely disagree; 2—I disagree; 3—I neither agree nor disagree; 4—I agree; and 5—I completely agree. If the experts chose response option 1 or 2, they were asked to make a comment or suggestion; their comments and suggestions were then reviewed until a consensus was reached. After the experts’ evaluation, the comments were organized into a table and analyzed by the research group regarding the coherence of the requested changes. All recommendations relevant to the methodology and purpose of the study were accepted. Minimal changes were made to the messages and no new content was added.

The experts’ responses were tabulated in Excel and exported to the software program IBM SPSS Statistics (Statistical Package for Social Sciences), version 25.0. The Content Validity Index (CVI) was adopted to measure the experts’ agreement regarding the validity of the educational technology’s content [48,49,50,51,52]. To determine the level of agreement among the experts, CVI scores were calculated using the number of experts who chose response options 4 or 5 for an item divided by the total number of experts who evaluated the item. The average of the CVI values of all the messages (S-CVI/Ave) and the content validity of the individual items (I-CVI) were calculated. An I-CVI greater than or equal to 0.78 was considered to indicate content validity [53,54]. The exact binomial distribution test indicated for small samples was performed, with a statistical significance level of *p* > 0.05 and a proportion of agreement of 0.95, to estimate the statistical reliability of the CVI.

## 3. Results

### 3.1. Preparation of Educational Messages

A total of 61 text messages were created with illustrations and divided into three themes: (1) presentation of the proposal and basic information about DM (*n* = 16); (2) physical exercise (*n* = 5); and (3) healthy eating (*n* = 40) (Table 1) (Appendix A).

### 3.2. Content Validation by Experts

A total of 623 experts were invited to participate in the validation process via email, with weekly reminders. Among them, 37 experts agreed to participate and completed the questionnaire. Regarding the experts’ demographics, 81% were female (*n* = 30), 32% were aged between 30 and 39 years old (*n* = 12), 43% were nurses (*n* = 16), 35% were nutritionists (*n* = 13), and 22% were physical education professionals (*n* = 8). Regarding their qualifications, 82% (*n* = 32) had a Ph.D. and 24% (*n* = 5) had a master’s degree. About 24.3% had worked in the profession for 0 to 9 years, 27% for 10 to 19 years, 21.6% for 20 to 29 years, and 27% for over 30 years. About 43% (*n* = 16) worked mainly in teaching and research. About 62% (*n* = 23) of the experts had published articles in the area of interest (Table 2).

Despite the fact that a CVI value above 0.80 (S-CVI/Ave) was obtained for all of the messages in terms of objective, content, and language (Table 3), the experts suggested some changes. Based on the input from the experts, the main changes requested were a review of the rules of the Portuguese language; rewriting of some sentences; replacement or removal of words and/or images that could confuse or improve the understanding of the content among older adults; and highlighting of certain words in bold.

## 4. Discussion

The instant messages, featuring educational content related to glycemic control, were prepared based on the validated content related to type 2 diabetes mellitus and healthy eating published by the Brazilian Association for the Study of Obesity and Metabolic Syndrome, the Brazilian Diabetes Society, and the Brazilian Ministry of Health [10,35,36,37,38,39,40,41,42]; the aim was to facilitate access to the information presented in these materials by older adults. The instant messages were evaluated by professionals in nursing, nutrition, and physical education. These educational messages were produced according to the preferences of older adults, taking into account the complexity of the information, text format, illustrations, and accessibility [55]. The proposed content aimed to promote adequate nutrition and healthy lifestyle habits.

The instant educational messages were written using simple, objective, and easy-to-understand language that is appropriate for the educational and cultural level of most older Brazilian adults, to ensure that the information is transmitted in a clear and accessible manner, including among those with low levels of education. Adapting the language to the specific needs of this group is essential for ensuring that the guidelines are not only understood, but also applied in everyday life. The texts were arranged in short sentences, with an easy-to-read font and a large font size [55]. Illustrations were used to facilitate understanding of the text. The use of illustrations supports the inclusion of population diversity in relation to race, sex, and body composition. Furthermore, using illustrations can attract older adults [56].

Instant messaging is available online due to the ease of storing information and the possibility of easy access to content when needed [55]. For example, the PronutriSenior Project, carried out in Portugal, showed that information about nutrition and health is essential for empowering older adults living in the community regarding their food choices, and these older adults have a preference for receiving audiovisual material with images [57].

Older adults are adapting to the digital age and are interested in online health-related information, especially after the COVID-19 pandemic. The challenge is to gather reliable information for the older population [58]. To ensure greater accuracy in the validation of educational messages, experts with experience in the subject were selected. This process was essential for ensuring that the evaluation was carried out by professionals with relevant knowledge. Even though the messages were evaluated as adequate, a qualitative analysis was carried out to examine the subjective recommendations of the experts; the experts’ recommendations were accepted and changes were made to make the messages more didactic and inviting.

Health education is essential for empowering people, making them protagonists in the management of their own condition. However, information provided through traditional, in-person lectures may not be appropriate for older adults who face physical or geographic challenges, such as mobility limitations or transportation difficulties. This can make it difficult to participate in events that require traveling, especially for those who live in remote or hard-to-reach areas, a reality observed in some Brazilian regions. Considering these limitations, seeking alternatives, such as digital technologies, is important. In Australia, participants over 55 years old stated that they felt comfortable using smartphones and sought reliable, up-to-date health information that is specific to the older population and allows for easy sharing with others [59].

Technology plays a crucial role in strengthening the guidance of health professionals and enabling the acquisition of knowledge, skills, and responsibility to effect attitude changes and increase decision-making power. This integrated approach can promote more effective health management and encourage self-care, which is essential for successful glycemic control as it can help patients avoid risky behaviors [60,61].

Health professionals who care for older adults, especially those with chronic diseases, should be familiar with health technologies that allow patients to take better care of themselves, be active in their health care, and improve their quality of life. This study considers messaging as an artifact that mediates the relationship between health professionals and the target audience, as well as a part of health education, which should be a dialogical practice, with efficient communication to stimulate autonomy and generate changes in risk behaviors; such education should not be prescriptive, imposing, or verticalized. Thus, messages should allow for co-responsibility of care, as they value the role of the patient in the health and disease process [62,63].

As positive points, we highlight the fact that the educational messages were evaluated by a multidisciplinary team and were validated in just one round of evaluation, with an average CVI score above 0.80. This result indicates that we have developed a high-quality educational technology to be used with older people diagnosed with DM. Instant educational messages validated by healthcare professionals with experience in the subject area go beyond traditional education and can reinforce the importance of adopting healthy lifestyle habits in order to reduce complications from the disease.

We highlight some limitations of our study that are present despite the methodological rigor. This study may be restricted to a specific audience, which may hinder the generalization of the results to a larger population. The experts selected to validate the educational technology might have differed in their interpretations of or responses to the validation questionnaire, and this subjectivity during the evaluation process might be influenced by the professional and cultural experiences of each individual. These limitations must be considered when designing new studies, to ensure that results are representative and valid.

## 5. Conclusions

Sixty-one text messages supported by illustrations, to be delivered via an instant messaging application, were developed to help older adults manage type 2 DM. During the process of validation by experts, these messages were considered relevant and clear for older adults and could be applied in the digital environment, with the objective of helping older adults manage type 2 DM.

The use of these messages could prove that educational actions through digital media can improve people’s glycemic control, as well as promote changes in eating patterns and healthy lifestyle habits, without the need to travel frequently for in-person appointments with healthcare professionals.

Finally, we emphasize the importance of conducting a pilot study to evaluate how these messages are received and what factors may influence their interpretation by the older population. Furthermore, there is a need to carry out clinical trials to verify the effectiveness of these messages in glycemic control among patients with diabetes. This study can serve as a model for other digital health initiatives, expanding the application of instant messaging to different chronic conditions.

## Figures and Tables

**Table 1 ijerph-22-00282-t001:** Theme, content, and quantity of educational messages.

Theme	Content of Educational Messages	Number of Items
1	Presentation of the proposal and basic information on type 2 DM.	9
	Information on body weight control and anthropometry.	7
2	Guidance on physical exercise.	4
3	Information on the importance of healthy eating and consuming natural, minimally processed, processed, and ultra-processed foods.	7
	Information on the consumption of sugar and sweeteners, as well as diet, light, and zero-calorie foods and drinks.	5
	Information on cardioprotective nutrition and the hygiene of fruits and vegetables.	3
	Information about foods that should be consumed in moderation, consumed in smaller quantities, and avoided.	4
	Information on food proportions and number of meals per day.	4
	Information about hypoglycemia and its treatment.	5
	General tips related to meal preparation.	10
	General tips for making the most of your meals.	3

**Table 2 ijerph-22-00282-t002:** Characterization of the specialists who participated in the validation of the educational instant messages.

Variable	*N* (37)	%
Sex		
Male	7	18.9
Female	30	81.1
Age range (years)		
20–29	2	5.4
30–39	12	32.4
40–49	10	27.0
50–59	7	18.9
60+	6	16.2
Career		
Physical education professional	8	21.6
Nurse	16	43.2
Nutritionist	13	35.1
Qualification		
Master’s degree	5	13.5
Ph.D.	32	86.5
Time of professional experience (years)		
0–9	9	24.3
10–19	10	27.0
20–29	8	21.6
30+	10	27.0
Professional experience		
Assistance	4	10.8
Teaching	3	8.1
Management	2	5.4
Research	2	5.4
Teaching and administration	1	2.7
Teaching and research	16	43.2
Teaching, assistance, and administration	1	2.7
Teaching, research, and administration	3	8.1
Teaching, research, and assistance	4	10.8
Research, assistance, and management	1	2.7
Published article on diabetes mellitus		
Yes	23	62.2
No	14	37.8

**Table 3 ijerph-22-00282-t003:** Questionnaire regarding validation of educational messages completed by experts.

Variables		I-CVI *
1	Objective	0.84
1.1	The contents meet the needs of people with DM2.	0.86
1.2	The contents are useful for improving self-care for people with DM2.	0.89
1.3	The contents can influence changes in ideas, behavior, and attitude.	0.81
1.4	It is suitable for sharing in the scientific community of the area.	0.78
1.5	It meets the objectives of healthcare professionals working with people with T2DM.	0.86
2	Structure and presentation	0.85
2.1	The information is presented clearly and objectively.	0.81
2.2	The information is scientifically correct.	0.81
2.3	There is a logical sequence in the proposed topics.	0.95
2.4	The information is written according to the rules of the Portuguese language.	0.92
2.5	The writing is appropriate for the proposed target audience.	0.84
2.6	The font size is adequate.	0.81
2.7	The illustrations are clear and sufficient.	0.86
2.8	The number of messages is adequate for use over twelve weeks.	0.78
3	Relevance	0.86
3.1	The topics cover the key aspects that need to be emphasized.	0.84
3.2	The messages encourage the construction of knowledge for self-care.	0.81
3.3	The messages include the topics necessary to build people’s knowledge about DM2.	0.89
3.4	The messages are appropriate for use by any healthcare professional.	0.89
Total CVI		0.85

Source: authors’ data, 2024. Legend: I-CVI: content validity of individual items; DM2: type 2 diabetes mellitus; * statistical significance with *p* > 0.05 according to binomial test.

## Data Availability

The raw data supporting the conclusions of this article will be made available by the authors upon request.

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
