# Peer review of "Connecting Health and Technology: Validation of Instant Messaging for Use as Diabetes Mellitus Control Strategy in Older Brazilian Adults"

_ijerph, 2025, doi:10.3390/ijerph22020282_

Round 1
Reviewer 1 Report
Comments and Suggestions for Authors
Costa et al has performed remarkable study on the Connecting Health and Technology: Validation of Instant Messaging for Use as Diabetes Mellitus Control Strategy in Brazilian Older Adults. However, they must attend the comments below before recommending acceptance.
i. Authors should enrich their introduction section with symptoms, health risk and complications associated with diabetes mellitus
ii. In the methodology section, authors should include the age range and occupation of the target respondents considered for the study
iii. Also, their level of literacy and sex of the respondents should be included in the methodology section
iv. Authors should carefully discuss the important findings of their results. This is lacking in the discussion section.
Comments on the Quality of English LanguageThe use of English can be improved
Author Response
Thank you very much for taking the time to review this manuscript. Please find the detailed responses below and the corresponding revisions/corrections highlighted/in track changes in the re-submitted files.
Comments 1: i. Authors should enrich their introduction section with symptoms, health risk and complications associated with diabetes mellitus
Response 1: Thank you for pointing this out. We add information in the line 46 – 54
Comments 2: ii. In the methodology section, authors should include the age range and occupation of the target respondents considered for the study
Response 2: For the selection of specialists, the age variable was not considered as an inclusion criterion, as the Fehring criterion points to master's and doctorate degrees, which already require a longer period of time in the profession. The occupation of specialists is entered in the line 160.
Comments 3: iii. Also, their level of literacy and sex of the respondents should be included in the methodology section
Response 3: For the selection of experts, the variable gender and literacy were not considered as an inclusion criterion. However, to be able to participate in the validation, the participant had to reach a minimum score in the Fehring classification, which takes into account educational level. The educational level of the specialist is included on the table 2.
Comments 4: iv. Authors should carefully discuss the important findings of their results. This is lacking in the discussion section.
Response 4: Thank you for pointing this out. We agree with this comment. We add information in the line 288
Reviewer 2 Report
Comments and Suggestions for Authors
The main intention of the study is to create instant alerts for diabetes patients, helping them maintain their health status and stay fit through the administration of healthy foods. The survey conducted among diabetes patients of different age groups may provide valuable insights into managing diabetes effectively. However, there are certain revisions needed to improve the manuscript’s quality from the readers' perspective:
-
Why have the authors mentioned only WhatsApp as a communication platform, rather than other social networks? People use various social media platforms for sharing messages, and the choice of WhatsApp may need further justification.
-
Social networks not only share authenticated information but also sometimes spread false information. What mitigation strategies do the authors propose to address this issue? It would be helpful to mention them.
-
A significant portion of Brazilian older adults do not use mobile phones due to financial and health-related reasons. Given this, how can the authors claim that technology is highly effective in connecting health-related information to this demographic?
-
Could the authors clarify how many people received the educational messages via WhatsApp? What is the exact population size that was reached with these messages?
-
Diabetes patients often experience polyphagia (insatiable hunger), and they may not understand why they are eating excessively. In this context, do the authors believe that these educational messages can help reduce the severity of diabetes, particularly in managing this symptom?
The quality of the language can be improved further to improve the standard of the journal.
Author Response
Thank you very much for taking the time to review this manuscript. Please find the detailed responses below and the corresponding revisions/corrections highlighted/in track changes in the re-submitted files.Comments 1: Why have the authors mentioned only WhatsApp as a communication platform, rather than other social networks? People use various social media platforms for sharing messages, and the choice of WhatsApp may need further justification.
Response 1: We emphasize the use of WhatsApp because it is the most used social network in Brazil, including by the elderly population. However, educational messages can also be shared through other social networks depending on the target population. The justification for choosing WhatsApp is inserted in the line 99.
Comments 2: Social networks not only share authenticated information but also sometimes spread false information. What mitigation strategies do the authors propose to address this issue? It would be helpful to mention them.
Response 2: Messages validated by health professionals have greater credibility with the population and help reduce the spread of false information. Furthermore, the study recommends that these messages be sent by primary health care professionals, serving as support and assisting in the care of elderly people with DM. We insert the information in the line 288.
Comments 3: A significant portion of Brazilian older adults do not use mobile phones due to financial and health-related reasons. Given this, how can the authors claim that technology is highly effective in connecting health-related information to this demographic?
Response 3: Thank you for pointing this out. We agree with this comment. We add information in the line 107.
Comments 4: Could the authors clarify how many people received the educational messages via WhatsApp? What is the exact population size that was reached with these messages?
Response 4: This article aimed to create and validate, by experts, the educational technology “educational messages”. This educational technology will be used in a randomized clinical trial by our research group. However, the idea of this article is to publish educational messages, already validated by experts, so that they can be sent to the elderly population by health professionals.
Comments 5: Diabetes patients often experience polyphagia (insatiable hunger), and they may not understand why they are eating excessively. In this context, do the authors believe that these educational messages can help reduce the severity of diabetes, particularly in managing this symptom?
Response 5: Yes, the authors believe that these educational messages can contribute to reducing the severity of DM, especially in controlling polyphagia. By providing clear information about eating foods that promote satiety, appropriate meal frequency and the importance of hydration, these messages help patients better understand and manage this symptom. Furthermore, guidance from health professionals, such as primary health care nutritionists, increases the credibility of the information and can encourage behavioral changes that are beneficial for glycemic control.
Reviewer 3 Report
Comments and Suggestions for Authors
General Comment:
There are readability issues in the entire paper. Extensive language corrections are required. Please seek professional help in this regard.
Line 48-49 Authors mentioned, “Brazil is 48 in 6th place….” It would be better to mention the current prevalence of DM in Brazil. According to previously conducted studies, what is the picture of glycemic control in the Brazilian DM population?
Line 82-83 “It is part of the Brazilian lifestyle used by 147 mil- 82 lion Brazilians, of which 86% use the application daily. “The authors should highlight, Does this statistic include all age groups? Generally, young people use social media a lot. Is there any statistic available for older adults?
Line 84: “Older adults also use the tool to clarify doubts about health care (25)” If e-health literacy has been evaluated in older adults in Brazil, authors should add the results and recent statistics.
Line 131: I wonder why the authors didn’t mention how translation was performed, was it done by language experts? It should be written.
Line 167: “The Content Validity Index (CVI)” authors should please justify why this method was chosen instead of others (like Cohen’s kappa, Tinsley-Weiss T index, etc.)
Line 186: “Regarding the expert's profile, 81% were female…”I suggest the information regarding the experts’ profile would be more appealing to and easy to understand for the reader if presented in the form of a table.
Line 198: “Table 1. Validation questionnaire for educational messages by expert experts.” I should be Table 2. The previous table was also labelled as Table 1.
Also, the authors wrote “expert experts.”. I suppose it’s a “ Validation questionnaire for educational messages by experts.
Table 1. In the validation questionnaire for educational messages by experts. Under objective “3 Language, “The heading “Language” does not match with the points below (3.1 to 3.4).
Line 241: “such as digital technologies”, How is internet coverage in remote and rural areas in Brazil? It can be discussed here as it would impact the reach of this product.
Line 252: I wonder about the strengths and limitations of the study not discussed. Authors must consider this point.
Line 260: In the conclusion, recommendations can be added such as pilot testing, to further assess clarity and comprehensibility.
Author Response
Thank you very much for taking the time to review this manuscript. Please find the detailed responses below and the corresponding revisions/corrections highlighted/in track changes in the re-submitted files.
Comments 1: There are readability issues in the entire paper. Extensive language corrections are required. Please seek professional help in this regard.
Response 1: Thank you for pointing this out. We hire the English proofreading service recommended by MDPI.
Comments 2: Line 48-49 Authors mentioned, “Brazil is 48 in 6th place….” It would be better to mention the current prevalence of DM in Brazil. According to previously conducted studies, what is the picture of glycemic control in the Brazilian DM population?
Response 2: Thank you for pointing this out. We add information in the line 57 and line 64.
Comments 3: Line 82-83 “It is part of the Brazilian lifestyle used by 147 mil- 82 lion Brazilians, of which 86% use the application daily. “The authors should highlight, Does this statistic include all age groups? Generally, young people use social media a lot. Is there any statistic available for older adults?
Response 3: Thank you for pointing this out. We add information in the line 94.
Comments 4: Line 84: “Older adults also use the tool to clarify doubts about health care (25)” If e-health literacy has been evaluated in older adults in Brazil, authors should add the results and recent statistics.
Response 4: Thank you for pointing this out. We add information in the line 101 and 105.
Comments 5: Line 131: I wonder why the authors didn’t mention how translation was performed, was it done by language experts? It should be written.
Response 5: Thank you for pointing this out. We hire the English proofreading service recommended by MDPI
Comments 6: Line 167: “The Content Validity Index (CVI)” authors should please justify why this method was chosen instead of others (like Cohen’s kappa, Tinsley-Weiss T index, etc.)
Response 6: The Content Validity Index (CVI) was adopted to measure experts' agreement regarding the validity of the content of educational technology as it is widely used, easy to apply and interpret. We added new references that used CVI in the line 189
Comments 7: Line 186: “Regarding the expert's profile, 81% were female…”I suggest the information regarding the experts’ profile would be more appealing to and easy to understand for the reader if presented in the form of a table.
Response 7: Thank you for pointing this out. We add information on a table in the line 215
Comments 8: Line 198: “Table 1. Validation questionnaire for educational messages by expert experts.” I should be Table 2. The previous table was also labelled as Table 1. Also, the authors wrote “expert experts.”. I suppose it’s a “ Validation questionnaire for educational messages by experts.
Response 8: Thank you for pointing this out. We have changed.
Comments 9: Table 1. In the validation questionnaire for educational messages by experts. Under objective “3 Language, “The heading “Language” does not match with the points below (3.1 to 3.4).
Response 9: We identified the mistake and made the correction, instead of language the topic addresses relevance (line 224). Language is addressed in items 2.4, 2.5 and 2.6.
Comments 10: Line 241: “such as digital technologies”, How is internet coverage in remote and rural areas in Brazil? It can be discussed here as it would impact the reach of this product.
Response 10: Thank you for pointing this out. We add information in the line 93.
Comments 11: Line 252: I wonder about the strengths and limitations of the study not discussed. Authors must consider this point.
Response 11: Thank you for pointing this out. We agree with this comment. We add information in the line 288
Comments 12: Line 260: In the conclusion, recommendations can be added such as pilot testing, to further assess clarity and comprehensibility.
Response 12: Thank you for pointing this out. We agree with this comment. We add information in the line 313